# Flatness-guided hyper-parameter optimization

### Abstract

We propose a novel white-box approach to hyper-parameter optimization. Motivated by recent work establishing a relationship between flat minima and generalization, we first establish a relationship between the Hessian norm and established sharpness metrics. Based on this, we seek to find hyper-parameter configurations that improve flatness by minimizing the upper bound on the sharpness of the loss. By using the structure of the underlying neural network, we derive semi-empirical estimates for the sharpness of the loss, and attempt to find hyper-parameters that minimize it in a randomized fashion. Through experiments on 14 classification datasets, we show that our method achieves strong performance at a fraction of the runtime.

## 1 Introduction

A typical machine learning pipeline involves using a combination of processes that have hyper-parameters that the analyst sets. There is significant interest in automatically computing a Pareto-optimal set of hyper-parameters tailored to the problem (Agrawal et al. (2019); Cowen-Rivers et al. (2022); Li et al. (2017); Bergstra et al. (2011); Bergstra and Bengio (2012); Falkner et al. (2018); Eriksson et al. (2019); Ansel et al. (2014b); Snoek et al. (2012); Hernández-Lobato et al. (2014); Swersky et al. (2014); Snoek et al. (2015); Bergstra et al. (2013)). In parallel, there is a venerable line of work studying the loss landscapes of neural networks (Hochreiter and Schmidhuber (1994; 1997); Hinton and Van Camp (1993); Chaudhari et al. (2019); Keskar et al. (2016); Dziugaite and Roy (2017); McAllester (1999); Neyshabur et al. (2014; 2017); Li et al. (2018); Seong et al. (2018); Dauphin et al. (2014); Choromanska et al. (2015); Zhang et al. (2021)). Notably, prior work has shown the effectiveness of improving the smoothness of loss surfaces via batch normalization (Santurkar et al. (2018)) and filter normalization (Li et al. (2018)).

Hyper-parameter optimization (HPO) is well-studied, with the most popular approaches being based on Bayesian optimization (Snoek et al. (2012); Hernández-Lobato et al. (2014); Swersky et al. (2014); Bergstra et al. (2013)), while other work suggests alternative approaches such as random search (Bergstra and Bengio (2012)) and tabu search (Agrawal et al. (2019)). Smith (2018) discusses empirical methods to manually tune hyper-parameters based on the performance of the current system. However, although HPO has repeatedly been shown to improve learner performance (Tantithamthavorn et al. (2016); Majumder et al. (2018)), much applied machine learning research either does not use HPO, or uses computationally expensive methods such as grid search. Some of this reluctance to use HPO stems from the general view that it is computationally expensive. For example, Tran et al. (2020) comment, "*Regardless of which hyper-parameter optimization method is used, this task is generally very expensive in terms of computational costs.*" Moreover, there is a growing concern to reduce the carbon emissions from ML experiments (Lacoste et al. (2019)).

Motivated by the need for computationally cheaper HPO methods, we pose the following question: *can we aim to directly improve the desirable properties of loss landscapes by exploiting the structure of the learning algorithm?* Specifically, recent work has repeatedly endorsed the relationship between the *flatness* of local minima and generalization ability of networks (Keskar et al. (2016); Jiang et al. (2019); Neyshabur et al. (2017); Dziugaite and Roy (2017); Li et al. (2018); Jastrzebski et al. (2017)). We use four major advances in the theoretical understanding of loss landscapes: (i) Wu and Su (2023) show that SGD can escape from low-loss, sharp minima (measured by the Frobenius norm of the Hessian) exponentially fast; (ii) Dauphin et al. (2014) used the line of work starting with Bray and Dean (2007) to show that saddle points are

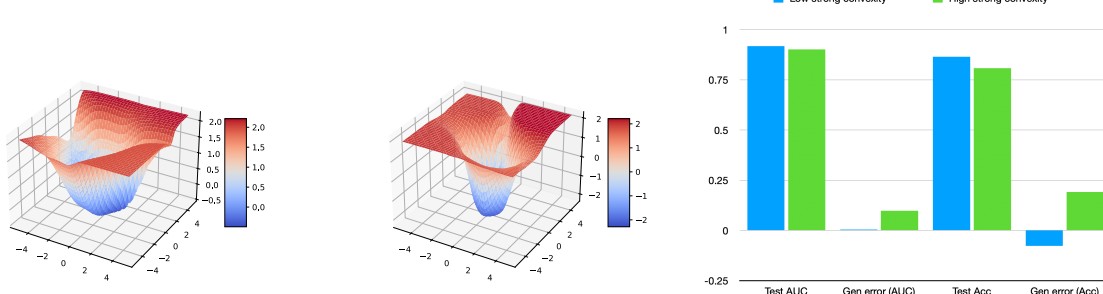

Figure 1: Landscapes (plotted using the method of Li et al. (2018)) with their corresponding metrics, on the Australian (binary classification, imbalanced) dataset. **Left:** a landscape with lower sharpness (0.112) and a wider minima. **Middle:** a landscape with higher sharpness (1.133). The sharpness values are computed using the result of Theorem 1. **Right:** Test metrics and generalization error for the two hyper-parameter configurations. Although the sharper configuration converged faster to a training error of 0, it generalizes poorly and performs worse on the test set.

exponentially more likely than local minima; (iii) gradient descent dynamics repel from saddle points (iv) the sharpness measure proposed by Keskar et al. (2016) have been repeatedly endorsed to correlate well with generalization (Jiang et al. (2019)). RbkEw:W2 We use these advances to develop a novel, computationally inexpensive HPO method that demonstrates strong performance. This continues a line of work studying efficient methods for HPO (Paul et al. (2019)). Through this work, we encourage researchers to adopt hyper-parameter optimziation in their applied ML work and avoid expensive and ineffective strategies such as random search and grid search.

We make the following connection between hyper-parameters and the sharpness of the loss surface. Consider the set of all loss functions. These loss functions depend on the data, the weights, and the hyper-parameters, so that for a fixed dataset, choosing a set of hyper-parameters materializes a subset of the neural network loss functions. Our method attempts to compute the lower bound on the sharpness for each set of loss functions, and materializes the neural networks corresponding to the 10 lowest values of sharpness.

In this work, we show that minimizing the supremum of the sharpness is equivalent in formulation to computing a lower bound on the Hessian norm of the loss in a mini-batch fashion. Next, we demonstrate a semi-empirical method of computing the sharpness of a loss function parameterized by the hyper-parameters of the model. The result we obtain is general enough to cover a wide range of network topologies. We use this result to motivate a hyper-parameter optimization method that uses the sharpness as a heuristic for search. Our method requires fewer full-length training runs of the learning algorithm, instead relying on one-epoch cycles to compute the sharpness, and discarding hyper-parameter configurations that are not promising.

Figure 1 shows the motivation for our approach. The left side shows a landscape with lower sharpness as computed by our method (and a flatter minima), which in turn has much lower generalization error for both accuracy and AUC[1]; the middle shows a landscape with a higher sharpness value, which led to a much higher generalization error for both metrics. Note that for the latter case, the training stopped early since both training AUC and accuracy reached 1; however, the model generalized poorly, and did worse on the test set. Similar plots and their corresponding performances on more datasets are shown in Appendix E.

Our contributions are as follows:

- We propose a novel white-box hyper-parameter optimization algorithm based on minimizing the sharpness of the loss, RbkEw:W1a which has been shown to negatively correlate with generalization performance (Section 3.2).

---

[1] RuHZU:W2a These are computed as $\text{metric}_{\text{train}} - \text{metric}_{\text{test}}$ .

- We derive semi-empirical estimates of the infimum of the Hessian norm, as a method of computing the sharpness. `RbkEw:W1b` Crucially, this does not rely on materializing the exact Hessian at runtime, which is computationally prohibitive (Section 3.3.

- We show that our algorithm achieves strong performance in HPO across 14 datasets at a fraction of the computational cost. `RbkEw:W1c` Compared to the fastest algorithm we compare against, our method is $\approx 200\%$ faster (Section 4.1).

To allow others to reproduce our work, our code is available online[2].

## 2    Related Work

This section briefly discusses related work; for a more comprehensive discussion, please see Appendix A.

**Hyper-parameter optimization.** In its general form, hyper-parameter optimization (HPO) solves the problem of finding a non-dominated hyper-parameter configuration under some budget. Early works (Bergstra and Bengio (2012); Bergstra et al. (2011)) showed the strength of random search, but since then, Bayesian Optimization has become increasingly popular. For example, the Tree of Parzen Estimators (TPE) algorithm models $p(x|y)$ using two kernel density estimates depending on whether $y$ is below or above some quantile, and optimizes the Expected Improvement (EI).

HyperBand (HB) (Li et al. (2017)) uses a procedure called "successive halving", which starts by randomly sampling a set of configurations and testing them under a limited budget, retaining only the best-performing ones and allocating those greater resources. At its core, its strategy is to aggressively prune poor-performing configurations so that more promising ones can be allocated more resources. Algorithms such as BOHB (Falkner et al. (2018)) and DEHB (Awad et al. (2021)) improve upon these in different ways: DEHB uses a distributed computing approach and combines differential evolution with HB, while BOHB combines a slightly modified version of the BO-based TPE with HB.

HEBO (Cowen-Rivers et al. (2022)) use a combination of input and output transformations along with NSGA-II to optimize a multi-objective acquisition function. TuRBO (Dou et al. (2023)) assumes the hyper-parameter to performance mapping is Lipschitz, and uses an ensemble of learners to predict performance, using the prediction to update its Gaussian Process model instead if that prediction is poor.

**Flat minima and generalization.** The connection between "flat minima" and generalization has been repeatedly endorsed. The flatness of minima has been defined in various ways, such as the volume of hypercuboids such that the loss is within a tolerance (Hochreiter and Schmidhuber (1997)) and as robustness to adversarial perturbations in weight space (Keskar et al. (2016)). The specific formulation of Keskar et al. (2016) is

$$\zeta(\boldsymbol{w}; \epsilon) = \frac{\max_{\boldsymbol{v} \in \mathcal{B}(\epsilon, \boldsymbol{w})} f(\boldsymbol{v}) - f(\boldsymbol{w})}{1 + f(\boldsymbol{w})}$$

This notion of sharpness was also endorsed by a large-scale study of complexity measures by Jiang et al. (2019). The relationship between flatness and generalization was later also endorsed by several works (Neyshabur et al. (2017); Li et al. (2018); Wu and Su (2023)). `RuHZU:W5` It is worth noting that flat minima are not a *necessary* condition for good generalization performance (Wen et al. (2023); Dinh et al. (2017)); however, Jiang et al. (2019) show a strong causal relationship between sharpness and generalization performance, which motivates our method. We defer the reader to Appendix A for a more detailed review.

At this point, we also distinguish this work from Sharpness-Aware Minimization (SAM) (Foret et al. (2020)). The latter attempts to find flat minima *during optimization*, i.e., once hyper-parameters have been chosen. As such, SAM can be used in conjunction with our method, and we do not compare against it.

---

[2]https://anonymous.4open.science/r/ahsc-hpo/

# 3 Method

## 3.1 Notation and Assumptions

For any learner, $X \in \mathcal{X}$ will represent the independent variables and $y$ will represent the labels; $W$ represents the weights of a neural network. We use $m$ to denote the number of training samples, $n$ to denote the number of features, $k$ to denote the number of classes, and $E$ to denote the loss function. $\boxed{\text{R6ZG2:W4a}}$ When we wish to express the loss function as parameterized by a set of hyper-parameters $H$, we will denote it as $E(\cdot; H)$ .

For a feedforward network, $L$ represents the number of layers, and at each layer, the following computation is performed: $z^{[l]} = W^{[l]T}a^{[l-1]} + b^{[l]}$ and $a^{[l]} = g^{[l]}(z^{[l]})$ where $a^{[0]} = X$. Here, $b^{[l]}$ is the bias vector at layer $l$, $W^{[l]}$ is the weight matrix at layer $l$, and $g^{[l]}$ is the activation function used at layer $l$.

We use $\nabla_W \cdot$ to denote the first gradient of $\cdot$, with respect to $W$. $\nabla^2_W \cdot$ denotes the Hessian, $\boxed{\text{R6ZG2:W5}}$ which is a rank-4 tensor obtained by computing the gradient of $E$ with respect to $W$ twice . Finally, $\mathcal{B}(\epsilon, w)$ denotes an $\epsilon-$ball centered at $w$. Whenever unspecified, the matrix norm is the Frobenius norm. We use $\mathbb{S}^n$ to denote the set of $n \times n$ real, symmetric matrices. We defer to Appendix B for background definitions in convex optimization.

$\boxed{\text{R6ZG2:W6a}}$ For notational convenience, we will use lb $f$ to denote *some lower bound* on a function $f$. Our method is based on obtaining lower bounds on sharpness for a chosen set of hyper-parameters, but it cannot be guaranteed that these lower bounds are the *greatest* lower bounds (i.e., the infimum). With some abuse of notation, when we write lb $f(x) = g$ for some expression $g$, it is equivalent to $f(x) \geq g \forall x \in \text{dom } f$.

**Assumption 1.** We assume the space $\mathcal{X}$ is Polish[3]. We use the Frobenius norm of the Hessian to define the sharpness of a loss landscape. In this context, completeness of the underlying metric space is necessary. Moreover, completeness is an important assumption in gradient descent, since it guarantees the existence of limits for Cauchy sequences. Finally, separability helps avoid issues with measurability when we use a covering argument in the next section.

**Asssumption 2.** We also assume the activation functions used in every layer satisfy $\frac{dg^{[l]}}{dx} \leq 1$. Most commonly used activations such as sigmoid, softmax, ReLU, and GELU satisfy this property.

## 3.2 FGHPO: Flatness-guided HPO

$\boxed{\text{RbkEw:W4}}$ Based on the relationship between flat minima and generalization, we develop a hyper-parameter optimization method that in a randomized fashion, explicitly searches for flatter losses. Specifically, we are interested in the problem:

$$\inf_{h \in \mathcal{H}} \zeta_h(w; \epsilon)$$

where $\zeta$ is the sharpness measure proposed by Keskar et al. (2016), discussed above, and the subscript denotes that the loss is parameterized by the hyper-parameter set $h$.

$\boxed{\text{RuHZU:W1a}}$ In this paper, we derive a lower bound on the norm of the Hessian to compute the sharpness of the loss landscapes. Specifically, we solve the problem:

$$\text{lb} \sup_{h \in \mathcal{H}} \sup_{x \subset X} \|\nabla^2 E(x; h)\|$$

This mini-batch version accounts for large datasets for which a mini-batch approach is necessary. Note that it is necessary to use an upper bound approximation at the mini-batch level: using a lower bound instead yields 0 a majority of the time, making the search ineffective. We can relate this to the sharpness measure from Keskar et al. (2016): this formulation corresponds to minimizing the upper bound on the sharpness over the entire $\epsilon-$ball. In Appendix B.1, we note that the standard covering argument bounds the deviation of this mini-batch approximation of the supremum from the true supremum.

We first show that this lower bound is directly related to the sharpness measure established by Keskar et al. (2016):

---

[3]A Polish space is a complete, separable, metrizable space.

$$\zeta(w) = \frac{\max\limits_{w' \in \mathcal{B}(\epsilon, w)} f(w') - f(w)}{1 + f(w)}$$

$$\overset{(i)}{\cong} \max_{w' \in \mathcal{B}(\epsilon, w)} f(w') - f(w)$$

$$\overset{(ii)}{\cong} \frac{\epsilon^2}{2}\left\|\nabla^2 f(w)\right\|_2 \leq \frac{\epsilon^2}{2}\left\|\nabla^2 f(w)\right\|_F$$

where (i) is because the training error is typically small in practice (Neyshabur et al. (2017)) and (ii) follows from a second-order Taylor expansion of $f$ around $w$, as done by Dinh et al. (2017). Therefore, we wish to minimize the sharpness, which is equivalent to the formulation above. Equivalently, we have (with $r$ denoting the rank of the Hessian)

$$\zeta(w) \cong \tfrac{\epsilon^2}{2}\left\|\nabla^2 f(w)\right\|_2 \geq \tfrac{\epsilon^2}{2\sqrt{r}}\left\|\nabla^2 f(w)\right\|_F \geq \tfrac{\epsilon^2}{2\sqrt{n}}\left\|\nabla^2 f(w)\right\|_F$$

Therefore, we have:

$$\frac{\epsilon^2}{2\sqrt{n}}\left\|\nabla^2 f(w)\right\|_F \leq \zeta(w) \leq \frac{\epsilon^2}{2}\left\|\nabla^2 f(w)\right\|_F$$

so that the sharpness and a scaled version thereof provide bounds on the sharpness, and minimizing the Hessian norm implies both the lower and upper bounds on the sharpness are lowered.

Algorithm 1 shows our overall approach. We first sample $N_1$ random configurations (line 2). For each of these configurations, we first train the model for one epoch to bring the weights closer to their final weights (line 6). Training for a single epoch provides a balance between the cost associated with training fully (which would provide a more accurate estimate for the sharpness), and not training at all (which provides a very poor estimate). We discuss this further in Section 4.2. We then compute the sharpness of the loss in a mini-batch fashion (lines 8-10). This sharpness is computed using the result of Theorem 1 below. Since we wish to minimize the sharpness, it is important that we (a) at the mini-batch level, compute a *lower bound* on the sharpness to obtain an estimate for the best-case scenario (line 9); (b) overall, look at the *highest* value across mini-batches, since that represents a best-effort estimate at the lower bound globally (also line 9). We aim to minimize that upper bound (line 15). Importantly, if the lower bound is 0 (implying there is no information), we discard that configuration (lines 11-13). We pick the configurations corresponding to the $N_2$ lowest values of sharpness as computed above, and train those models fully (lines 15-17). Finally, we return the best-performing configuration.

In Appendix C, Theorem 2, we show that if the loss is smooth and $\mu-$strongly convex[4], then

$$f(x_t) - f(x^*) \leq (1 - \alpha\mu)^t (f(x_0) - f(x^*))$$

where $\alpha$ is the learning rate. Since we have $\forall t, f(x_t) - f(x^*) \geq 0$,

$$f(x_{t+1}) - f(x_t) \leq (1 - \alpha\mu)^{t+1} (f(x_0) - f(x^*))$$
$$\leq \exp(-\alpha\mu(t+1)) (f(x_0) - f(x^*))$$

which implies exponentially decaying benefit as the number of epochs increases. On the other hand, Theorem 2 also implies that for vanilla gradient descent, the number of steps required for convergence is inversely proportional to the strong convexity, so that minimizing the latter implies a greater number of steps is required to converge (which increases the runtime). To reduce the impact of this, we use the Adam (Kingma and Ba (2014)) optimizer. We leave it to future work to explore additional strategies, such as large adaptive learning rates, which can also lead to flatter losses (Jastrzebski et al. (2017)).

---

[4]Note that the strong convexity can be defined as the lower bound on the Hessian norm, which is what we use to compute sharpness.

---

**Algorithm 1** FGHPO

---

1: **Input:** Number of configurations to sample $N_1$, number of configurations to run $N_2$. Defaults: $N_1 = 50, N_2 = 10$.
2: $\mathcal{H}_0 \leftarrow \text{RANDOM}(\mathcal{H}, N_1)$
3: $S \leftarrow \phi$ {Sharpness values}
4: $P \leftarrow \phi$ {Performance scores}
5: **for** config $h$ in $\mathcal{H}_0$ **do**
6:     Train for one epoch using $h$
7:     $\mu_{max} = -\infty$
8:     **for** mini-batch $x \subset X$ **do**
9:         $\mu_{max} = \max(\mu_{max}, \text{lb}\lVert \nabla^2 E(x; h)\rVert)$
10:     **end for**
11:     **if** $\mu_{max} > 0$ **then**
12:         S[h] $\leftarrow \mu_{max}$
13:     **end if**
14: **end for**
15: **for** config $h$ in $\text{LOWEST}(S, N_2)$ **do**
16:     P[h] $\leftarrow \text{RUN(h)}$
17: **end for**
18: **return** $\arg\max P$

---

### 3.3 Computing the sharpness

RuHZU:W1b We now derive a semi-empirical expression for the sharpness $(\text{lb}\lVert \nabla^2 E(x; h)\rVert)$ used in Line 9 of Algorithm 1. Our main result (Theorem 1) is that in the general multi-class classification setting with a softmax activation at the last layer,

$$\text{lb} \lVert \nabla_W^2 E \rVert = \frac{\lVert a_j^{[L-1]} \rVert}{\lVert W^{[L]} \rVert}$$

Importantly, the proof does not rely on the architecture of the network beyond the last two layers. That is, as long as the last two layers of the network are fully-connected, this theorem applies. RuHZU:W1c We first prove some auxiliary results that will be used in the proof for Theorem 1.

**Lemma 1** (Norm of Moore-Penrose pseudo-inverse)**.** *Let B be the left Moore-Penrose pseudo-inverse of some matrix A. Then*

$$\lVert B \rVert_2 = \frac{1}{\sigma_{min}(A)}$$

*where $\sigma_{min}(A)$ denotes the least singular value of A.*

*Proof.* It is well-established that $AB$ is the orthogonal projection to the column space of $A$, so that for every vector $y$, $ABy \perp (I - AB)y$. Therefore:

$$
\begin{aligned}
\|B\|_2 &= \max_{y \neq 0} \frac{\|By\|}{\|y\|} \\
&= \max_{y \neq 0} \frac{\|By\|}{\|ABy + (I - AB)y\|} \\
&= \max_{By \neq 0} \frac{\|By\|}{\|ABy + (I - AB)y\|} \\
&\leq \max_{By \neq 0} \frac{\|By\|}{\|ABy\|} \\
&\leq \max_{x \neq 0} \frac{\|x\|}{\|Ax\|} \\
&= \left( \min_{x \neq 0} \frac{\|Ax\|}{\|x\|} \right)^{-1} \\
&= \frac{1}{\sigma_{min}(A)}
\end{aligned}
$$

$\square$

**Lemma 2.** *For a deep learner with activations satisfying Assumption 2 in the hidden layers, the last two layers being fully-connected, and a softmax activation at the last layer,*

$$
\frac{\partial E}{\partial z_h^{[L]}} = a_h^{[L]} - \frac{1}{m} \sum_{i=1}^{m} [y^{(i)} = h]
$$

*under the cross-entropy loss.*

*Proof.* Consider the Iverson notation version of the general cross-entropy loss:

$$
E(\mathbf{a}^{[L]}) = -\frac{1}{m} \sum_{i=1}^{m} \sum_{j=1}^{k} [y^{(i)} = j] \log a_j^{[L]}
$$

Then,

$$
\begin{aligned}
\frac{\partial E}{\partial z_h^{[L]}} &= -\frac{1}{m} \sum_{i=1}^{m} \sum_{j=1}^{k} \frac{[y^{(i)} = j]}{a_j^{[L]}} a_j^{[L]} \left( \delta_{hj} - a_h^{[L]} \right) \\
&= -\frac{1}{m} \sum_{i=1}^{m} \sum_{j=1}^{k} [y^{(i)} = j] \left( \delta_{hj} - a_h^{[L]} \right) \\
&= -\frac{1}{m} \sum_{i=1}^{m} \sum_{j=1}^{k} [y^{(i)} = j] \delta_{hj} + \frac{1}{m} \sum_{i=1}^{m} \sum_{j=1}^{k} [y^{(i)} = j] a_h^{[L]} \\
&= -\frac{1}{m} \sum_{i=1}^{m} [y^{(i)} = h] + \frac{a_h^{[L]}}{m} \sum_{i=1}^{m} 1 \\
&= a_h^{[L]} - \frac{1}{m} \sum_{i=1}^{m} [y^{(i)} = h]
\end{aligned}
\tag{1}
$$

$\square$

**Theorem 1** (Sharpness for feedforward networks). *For a deep learner with activations satisfying Assumption 2 in the hidden layers, the last two layers being fully-connected, and a softmax activation at the last layer, the sharpness of the cross-entropy loss is lower bounded by* R6ZG2:W6b

$$\text{lb}\|\nabla^2_{W^{[L]}} E\| = \frac{\|a^{[L]}\|}{\|W^{[L]}\|} \tag{2}$$

*Proof.* From Lemma 2 and $z^{[L]} = W^{[L]} a^{[L-1]} + b^{[L]}$,

$$\nabla_{W^{[L]}} E = a^{[L]} a^{[L-1]T}$$

Now, we work toward the Hessian:

$$\frac{\partial^2 E}{\partial a^{[L]} \partial W^L} = I_{n^{[L]} \times n^{[L]}} \otimes a^{[L-1]T} + a^{[L]} \otimes \left( \frac{\partial a^{[L]}}{\partial a^{[L-1]T}} \right)^{-1}$$

which is of shape $(n^{[L]}, n^{[L]}, n^{[L-1]})$. Note that because the derivative inside the inverse is not square, this inverse is the Moore-Penrose pseudo-inverse. We have:

$$\frac{\partial a^{[L]}}{\partial a^{[L-1]}} = \frac{\partial a^{[L]}}{\partial z^{[L]}} \frac{\partial z^{[L]}}{\partial a^{[L-1]}}$$
$$= \left( \text{diag}(a^{[L]}) - a^{[L]} a^{[L]T} \right) W^{[L]} I_{n^{[L-1]} \times n^{[L-1]}}$$
$$\frac{\partial a^{[L]}}{\partial a^{[L-1]T}} = I_{n^{[L-1]} \times n^{[L-1]}} W^{[L]T} \left( \text{diag}(a^{[L]}) - a^{[L]} a^{[L]T} \right)$$

so that

$$\frac{\partial^2 E}{\partial a^{[L]} \partial W^L} = I_{n^{[L]} \times n^{[L]}} \otimes a^{[L-1]T} + a^{[L]} \left( I_{n^{[L-1]} \times n^{[L-1]}} W^{[L]T} \left( \text{diag}(a^{[L]}) - a^{[L]} a^{[L]T} \right) \right)^{-1}$$

Next,

$$\frac{\partial a^{[L]}}{\partial W^{[L]}} = \frac{\partial a^{[L]}}{\partial z^{[L]}} \frac{\partial z^{[L]}}{\partial W^{[L]}}$$
$$= \left( \text{diag}(a^{[L]}) - a^{[L]} a^{[L]T} \right) \otimes a^{[L-1]}$$

which is of shape $(n^{[L]}, n^{[L]}, n^{[L-1]})$. This yields the Hessian

$$\nabla^2_{W^{[L]}} E = \left( I_{n^{[L]} \times n^{[L]}} \otimes a^{[L-1]T} \right) \left( \left( \text{diag}(a^{[L]}) - a^{[L]} a^{[L]T} \right) \otimes a^{[L-1]} \right)^T +$$
$$a^{[L]} \left( I_{n^{[L-1]} \times n^{[L]}} W^{[L]T} \left( \text{diag}(a^{[L]}) - a^{[L]} a^{[L]T} \right) \right)^{-1} \left( \text{diag}(a^{[L]}) - a^{[L]} a^{[L]T} \right) \otimes a^{[L-1]}$$
$$\geq \frac{\|a^{[L]}\|}{\|W^{[L]}\|}$$

where for the last step we used Lemma 1, dropped the first term to get a lower bound, and used $\sigma_{min} X \leq \sigma_{max} X = \|X\|_2$. □

Table 1: Experimental results on various classification datasets. FGHPO is our method. Values shown are medians over 20 repeats. Statistically best results are highlighted in **bold** (see Section 4 for details).

| Image | | |
|---|---|---|
| **Dataset** | **HPO method** | **Accuracy** |
| MNIST | FGHPO | **98.65** |
| | Hyperopt | 97.22 |
| | Random | **98.95** |
| | TuRBO | **98.99** |
| | HEBO | **98.94** |
| | BOHB | 98.85 |
| SVHN | FGHPO | **86.63** |
| | Hyperopt | 67.20 |
| | Random | **91.86** |
| | TuRBO | 80.41 |
| | HEBO | 92.89 |
| | BOHB | 79.67 |

| Bayesmark | | |
|---|---|---|
| **Dataset** | **HPO method** | **Score** |
| breast | FGHPO | **93.98** |
| | Hyperopt | **92.68** |
| | Random | 90.45 |
| | TuRBO | 88.97 |
| | HEBO | 90.84 |
| | BOHB | 92.37 |
| digits | FGHPO | 84.74 |
| | Hyperopt | **96.85** |
| | Random | 87.39 |
| | TuRBO | 89.51 |
| | HEBO | **95.24** |
| | BOHB | 91.50 |
| iris | FGHPO | **91.00** |
| | Hyperopt | 79.83 |
| | Random | 82.98 |
| | TuRBO | 83.52 |
| | HEBO | 78.27 |
| | BOHB | **92.30** |
| wine | FGHPO | **92.35** |
| | Hyperopt | 83.98 |
| | Random | 76.59 |
| | TuRBO | 81.15 |
| | HEBO | 74.93 |
| | BOHB | 81.68 |

| OpenML | | |
|---|---|---|
| **Dataset** | **HPO method** | **AUC** |
| vehicle | FGHPO | **0.885** |
| | Hyperopt | **0.883** |
| | Random | 0.873 |
| | TuRBO | **0.882** |
| | HEBO | **0.884** |
| | BOHB | **0.883** |
| blood-transf... | FGHPO | 0.721 |
| | Hyperopt | **0.728** |
| | Random | 0.708 |
| | TuRBO | 0.720 |
| | HEBO | 0.718 |
| | BOHB | **0.725** |
| Australian | FGHPO | **0.934** |
| | Hyperopt | **0.932** |
| | Random | 0.928 |
| | TuRBO | **0.932** |
| | HEBO | **0.935** |
| | BOHB | **0.928** |
| car | FGHPO | 1.0 |
| | Hyperopt | **1.0** |
| | Random | 1.0 |
| | TuRBO | 1.0 |
| | HEBO | 1.0 |
| | BOHB | **1.0** |
| phoneme | FGHPO | 0.560 |
| | Hyperopt | **0.564** |
| | Random | 0.561 |
| | TuRBO | **0.564** |
| | HEBO | 0.563 |
| | BOHB | **0.563** |
| segment | FGHPO | **0.961** |
| | Hyperopt | **0.960** |
| | Random | 0.955 |
| | TuRBO | **0.960** |
| | HEBO | **0.961** |
| | BOHB | **0.960** |
| credit-g | FGHPO | 0.778 |
| | Hyperopt | **0.782** |
| | Random | 0.766 |
| | TuRBO | 0.763 |
| | HEBO | 0.763 |
| | BOHB | **0.752** |
| kcl | FGHPO | **0.816** |
| | Hyperopt | **0.817** |
| | Random | 0.769 |
| | TuRBO | 0.775 |
| | HEBO | **0.816** |
| | BOHB | **0.785** |

## 4 Experiments

We compare our approach based on the sharpness (which we call FGHPO) with other popular hyper-parameter optimization algorithms. We randomly sample 50 configurations, compute their sharpness (computed by Theorem 1), and run the top 10, reporting the best-performing one. We repeat all experiments 20 times, and compare results using pairwise Mann-Whitney tests (Mann and Whitney (1947)) with a Benjamini-Hochberg correction procedure for p-values (as endorsed by Farcomeni (2008)), employing a 5% significance level. `R6ZG2:W2` For all baselines, we use default settings with a budget of 50 evaluations. In all cases, models are trained for 100 epochs. `RuHZU:W8b` At test time, for the random and Bayesian Optimization (BO)-based methods[5], we use the most promising candidate produced by the 50 evaluations over the training set (based

---

[5]This is all the methods except ours and random.

on validation set performance at training time). For our method, we use the best performing configuration (based on validation set performance) over the 10 with the lowest sharpness.

For tabular datasets, we experiment on the Bayesmark datasets used in the NeurIPS 2020 Black-Box Optimization Challenge and the 8 datasets used in the MLP benchmarks in HPOBench (Eggensperger et al. (2021)). For convolutional networks, we run experiments on MNIST (LeCun (1998)) and SVHN (Netzer et al. (2011)).

`RuHZU:W8a` We use the default train/test splits provided for all datasets to remain consistent with prior work. For the image datasets, we scale each feature to $[0, 1]$. For the OpenML datasets, we normalize samples to have an $l_2$ norm of 1. We do not perform any preprocessing for the Bayesmark datasets.

Table 2: Summary of our results. Wins, ties, and losses are determined by the results of statistical significance tests.

| Algorithm | Wins | Ties | Losses | (Wins + Tie)% |
|---|---|---|---|---|
| FGHPO | 1 | 8 | 5 | 64.3 |
| `R6ZG2:W3a` Hyperopt | 0 | 10 | 4 | 71.4 |
| Random | 0 | 2 | 12 | 14.3 |
| TuRBO | 0 | 5 | 9 | 35.7 |
| HEBO | 0 | 6 | 8 | 42.9 |
| BOHB | 0 | 8 | 6 | 57.1 |

For Bayesmark, we use the default set of hyper-parameters for MLPs, which has a size of $134M^6$. For MNIST, we used Conv - MaxPooling blocks, followed by two fully-connected layers. For SVHN, we used Conv - BatchNorm - Conv - MaxPooling - Dropout layers, followed by a fully-connected layer, a dropout layer, and a final fully-connected layer. The range of hyper-parameters for all these models is shown in Table 3.

We use the metrics employed by prior work to ensure a fair and consistent evaluation. For Bayesmark datasets, we report the mean normalized score, which first calculates the performance gap between observations and the global optimum and divides it by the gap between random search and the optimum. For MNIST and SVHN, we use the accuracy score. For the OpenML datasets, we use the area under the ROC curve, since we found that many of them had notable class imbalances.

Our results, shown in Table 1, demonstrate that FGHPO is capable of achieving strong performance on most datasets. These results are summarized in Table 2. We declare an algorithm as "winning" on a dataset if it outperforms every other algorithm; it "ties" if any other algorithm's performance is not statistically significantly different.

`R6ZG2:W1` We also checked that the final validation loss is more strongly correlated with the flatness than with the validation loss after one epoch. Using data from all of the OpenML datasets, we computed the Spearman correlation coefficient. As predicted, the sharpness that we computed was negatively correlated with the final validation loss ($\rho = -0.14, p < 0.05$); moreover, the validation loss after one epoch was not correlated with the final validation loss ($\rho = 0.08, p = 0.23$).

## 4.1 Runtime

In practice, computing the sharpness is cheap. In detail, we train the network with the given hyper-parameters for one epoch and then use the equations derived to compute the sharpness in mini-batches. The one epoch of training moves the network weights closer to the final position, so that the measured loss criterion is more accurate than from a randomly initialized point.

On a machine with an Intel Cascade Lake CPU with 4 vCPUs and 23GB RAM and no GPU, where we ran our NeurIPS Black-Box Optimization Challenge experiments, we measured the cost of computing the sharpness over 15 runs with varying batch sizes. The median number of batches was 15.51 (442 samples), which took a median of 0.47 seconds. Therefore, it takes 0.03s/batch/config to compute the sharpness. For

---

[6]`https://github.com/uber/bayesmark/blob/master/bayesmark/sklearn_funcs.py`

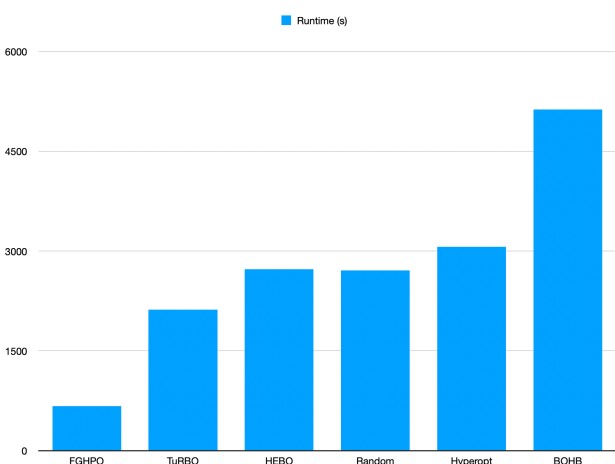

Figure 2: Algorithm runtimes on the vehicle dataset.

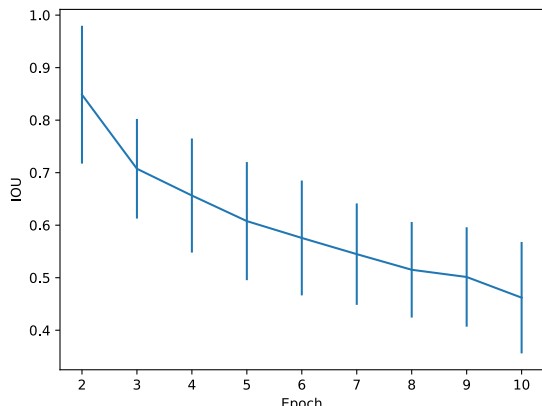

Figure 3: RuHZU:W9 Intersection over Union (IoU) of the set of the 10 configurations corresponding to the lowest sharpness, after one epoch and after epoch $i, i \in \{1, \ldots, 10\}$.

example, the breast cancer dataset has 569 samples. Using the mean batch size in the hyper-parameter space of 130, that evaluates to 4.38 batches, which we expect to take $0.03 \times 4.38 \times 50 = 6.57$s to compute the sharpness for 50 configurations.

The above experiments suggest that computing the sharpness this way is computationally cheap, since we train for the full epochs only for the top 10 configurations. Figure 2 shows the runtimes for each algorithm on the vehicle dataset, on a machine with an RTX 2060 Super. Our approach requires between 13% to 33% of the runtime compared to other algorithms.

## 4.2 The exploration-exploitation trade-off

We additionally tested how much the configurations corresponding to the 10 lowest sharpness values changed as the epochs progressed (since those are the only ones trained fully and evaluated). To do this, we computed the sharpness after each epoch, for 10 epochs, on the HPOBench datasets used in our paper. We then computed the IoU (Intersection Over Union) metric between the configurations in the top 10 after 1 epoch with those in the top 10 after 2, 3, ..., 10 epochs. As expected, this metric decreases gradually; after 10 epochs, this IoU has a mean of 47% (see Figure 3). This means that after 10 epochs, about half the configurations in the top 10 are the same as after epoch 1. We argue that this is an exploration-exploitation

tradeoff: by training for further epochs, we obtain a more accurate estimate, at additional computational expense. We maintain that the one-epoch training cycle balances the exploration-exploitation trade-off well, since it is not necessary for a landscape to have flat minima to generalize well (Wen et al. (2023); Dinh et al. (2017)).

## 5    Limitations

In this paper, we developed a novel white-box hyper-parameter optimization algorithm that, after some cheap computation, requires only 10 full runs to find a good configuration. We demonstrated our results on 12 tabular and two image datasets.

It has not escaped our attention that this method does not allow the user to specify a preference for evaluation metrics such as recall or precision. We leave this as future work. In particular, we exploit the fact that the Pareto frontier is a subset of the convex hull of the hyper-parameter performance scores. To find configurations that do well on some metric, we traverse the Pareto frontier and compute quantized[7] convex combinations of adjacent points and also test them. This is similar to the approach of Ammar (2004). However, this approach adds additional computational cost.

Our hyper-parameter optimization method has two key limitations: first, it is limited to learners for which a loss function can be defined. In some cases such as Naive Bayes, a surrogate such as the negative log-likelihood can be used, for which the sharpness can be computed. Even in cases where the loss is not twice-differentiable, one can use a finite difference approximation to compute the Hessian (see Nocedal and Wright (1999)):

$$\frac{\partial^2 f}{\partial x_i \partial x_j}(x) \approx \frac{1}{\epsilon^2}\left(f(x + \epsilon e_i + \epsilon e_j) - f(x + \epsilon e_i) - f(x + \epsilon e_j) + f(x)\right)$$

where the error is $\mathcal{O}(\epsilon)$. The second, potentially more important limitation is that the sharpness cannot be compared across learners, especially if different losses are used. For example, while algorithms such as TPE can be used on hyper-parameter spaces with multiple classes of learners, our approach cannot: the entire hyper-parameter space must have comparable sharpness values, for which the same class of learners (such as neural networks, Naive Bayes, logistic regression, etc.) must be used. However, this can be resolved by using the same loss function across learning algorithms. For example, a negative log-likelihood ratio loss has been proposed for neural classifiers (Yao et al. (2020)), which is compatible with other learners.

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

## A   Related Work

**Hyper-parameter optimization.** There is significant prior work in hyper-parameter optimization (Agrawal et al. (2019); Cowen-Rivers et al. (2022); Li et al. (2017); Bergstra et al. (2011); Bergstra and Bengio (2012); Falkner et al. (2018); Eriksson et al. (2019); Ansel et al. (2014b)). Indeed, as learning systems become more intricate, it is crucial that we eke out the most performance. However, this is a non-trivial problem, as evidenced by the long line of research in this direction.

The simplest form of hyper-parameter search is random search, which tries $n$ randomly chosen hyper-parameter configurations. Opentuner (Ansel et al. (2014a)) is a multi-armed bandit meta-technique with a sliding window that incorporates an exploration/exploitation trade-off based on the number of times a specific technique is used. It combines DE, a greedy bandit mutation technique, and hill-climbing methods.

Bayesian Optimization (BO) has emerged as the most popular technique for HPO. Bergstra et al. (2011) propose the Tree of Parzen Estimators (TPE) algorithm. Rather than model $p(y|x)$, TPE models $p(x|y)$ as

$$p(x|y) = \begin{cases} l(x) & y < y^* \\ g(x) & y \geq y^* \end{cases}$$

where $y^*$ is chosen so that $p(y < y^*) = \gamma$ for some quantile $\gamma$. The functions $l(x)$ and $g(x)$ are kernel density estimates. TPE optimizes the EI, which they show is equivalent to maximizing $l(x)/g(x)$. Snoek et al. (2012) use Gaussian Process (GP) models as the surrogate function in Bayesian optimization. They use Expected Improvement (EI) as the acquisition function. Similarly, Hernández-Lobato et al. (2014) use predictive entropy search (PES) as the acquisition function. Swersky et al. (2014) exploit iterative training procedures in their Bayesian optimization framework, which they call freeze-thaw Bayesian optimization. Snoek et al. (2015) use neural networks for modeling distributions over functions that yields an approach that scales linearly over data size (rather than cubically as in GP-based Bayesian optimization). BOHB (Falkner et al. (2018)) combines the BO-based TPE with HyperBand (Li et al. (2017)), replacing the initial random configurations with a model-based search. Notably, BOHB uses a single multi-dimensional KDE instead of hierarchical single-dimensional KDEs used by TPE. The authors of HEBO (Cowen-Rivers et al. (2022)) note that (i) even simple HPO problems can be non-stationary and heteroscedastic (ii) different acquisition functions can conflict. To tackle the former, they use the Box-Cox (Box and Cox (1964)) and Yeo-Johnson (Yeo and Johnson (2000)) output transformations and the Kumaraswamy (Kumaraswamy (1980)) input transformation. It also uses NSGA-II to optimize a multi-objective acquisition function. TuRBO (Dou et al. (2023)) assumes the hyper-parameter to performance mapping is Lipschitz, and generates pseudo-points to improve convergence of vanilla BO. It also uses an ensemble of learners to predict performance, and if the prediction is poor, uses it to instead update the GP model. Finally, we mention PriorBand (Mallik et al. (2023)), which incorporates an expert's prior beliefs about good configurations, but maintains good performance even if that prior is bad.

We defer to (Feurer and Hutter (2019)) and (Bischl et al. (2023)) for recent reviews on hyper-parameter optimization techniques. There is also a long line of work studying neural architecture search, which aims to find optimal architectures for a dataset. We refer the reader to White et al. (2023) for a comprehensive review of the field.

Several benchmarks have been proposed for hyper-parameter optimization: notable ones include YAHPO Gym (Pfisterer et al. (2022)), HPO-B (Arango et al. (2021)), and HPOBench (Eggensperger et al. (2021)).

**Flat minima and generalization.** The idea of flat minima was first studied by Hochreiter and Schmidhuber (1994). In particular, they define "flat minima" as large connected regions where the weights are $\epsilon-$optimal. Hochreiter and Schmidhuber (1997) intuit that because sharper minima require higher precision, flatter minima require less bits to describe. They use this intuition to show that flat minima correspond to minimizing the number of bits required to describe the weights of a neural network. This notion of minimum description length (MDL) (Rissanen (1983); Grünwald (2007)) was also exploited early on by Hinton and Van Camp (1993). Flat minima were revisited by Chaudhari et al. (2019), who noted that minima with low generalization error have a large proportion of their eigenvalues close to zero. They then construct a modified Gibbs distribution corresponding to an energy landscape $f$, and minimize the negative local entropy of this modified

distribution, and approximate the gradient via stochastic gradient Langevin dynamics (SGLD) (Welling and Teh (2011)). However, their assumptions were, admittedly unrealistic. Keskar et al. (2016) show that when using large batch sizes, optimizers converge to sharp minima, which are characterized by many large positive eigenvalues of the Hessian. Further, they define the notion of sharpness as a generalization measure as the robustness to adversarial perturbations in the parameter space:

$$\zeta(\boldsymbol{w}; \epsilon) = \frac{\max\limits_{|\boldsymbol{v}| \leq \epsilon(|\boldsymbol{w}|+1)} f(\boldsymbol{w} + \boldsymbol{v}) - f(\boldsymbol{w})}{1 + f(\boldsymbol{w})} \tag{3}$$

and compute this using 10 iterations of L-BFGS-B (Byrd et al. (1995)) with $\epsilon = \{10^{-3}, 5 \cdot 10^{-4}\}$. In particular, the above is closely related to the largest eigenvalue of $\nabla^2 f(\boldsymbol{w})$. This notion of sharpness was also endorsed by Jiang et al. (2019), who performed a large-scale study of many complexity measures on two datasets, with 2,187 convolutional networks. In particular, they endorse the following metrics for generalization: (i) variance of gradients (ii) squared ratio of magnitude of parameters to magnitude of perturbation, à la Keskar et al. (2016) (iii) path norm (Neyshabur et al. (2017)) (iv) VC-dimension (inversely correlated).

A line of work in physics (Baldassi et al. (2015; 2016)) showed that in the discrete weight scenario (a much more difficult problem), isolated minima were rare, but there existed accessible, dense regions of subdominant minima, and that these were robust to perturbations and generalized better. These authors devised algorithms explicitly designed to search for nonisolated minima. In the continuous weight space, nonisolated minima correspond to flat minima. Dziugaite and Roy (2017) obtain nonvacuous generalization bounds for deep overparameterized neural networks using the PAC-Bayes framework (McAllester (1999)). Neyshabur et al. (2014) showed that increasing the number of hidden units (which in turn, increases the number of trainable parameters) can lead to a decrease in generalization error with the same training error. Neyshabur et al. (2017) showed that sharpness as computed by (3) is not sufficient to capture the generalization behavior (but noting that "combined with the norm, sharpness does seem to provide a capacity measure"), and advocate for expected sharpness in the PAC-Bayesian framework, similar to Dziugaite and Roy (2017). They show that plots of expected sharpness versus KL divergence in PAC-Bayes bounds for varying dataset sizes capture generalization well. Li et al. (2018) showed that the sharpness of the loss surface correlates well with the generalization error. In their seminal paper, Jastrzebski et al. (2017) showed that SGD is a Euler-Maruyama discretization of a stochastic differential equation whose dynamics are influenced by the ratio of learning rate to batch size (which they call "stochastic noise"), and that SGD finds wider minima with higher stochastic noise levels than sharper minima. Wu and Su (2023) study the flat minima hypothesis through the lens of dynamical stability, and show that SGD will escape from overly sharp (measured by the Frobenius norm of the Hessian), low-loss areas exponentially fast. We note that they use the associate empirical Fisher matrix (AEFM) as an approximation for the Hessian, which holds for low empirical risk (and converges to the Hessian, see Kunstner et al. (2019)).

In search of flatter loss surfaces, Seong et al. (2018) propose the use of non-monotonic learning rate schedules. They advocate for large learning rates, which enable the optimization algorithm to escape sharp minima, and descend into flatter minima. The seminal works of Dauphin et al. (2014) and Choromanska et al. (2015) showed both theoretically and empirically that local minima are more likely to be located close to the global minimum. In particular, Dauphin et al. (2014) showed using the perspectives of random matrix theory (via the eigenvalue distribution of Gaussian random matrices (Wigner (1958))), statistical physics (via the analysis of critical points in Gaussian fields by Bray and Dean (2007)), and neural network theory (Saxe et al. (2013)) that saddle points are exponentially less likely than local minima.

Most recently, Wen et al. (2023) showed that sharpness is neither necessary, nor sufficient for generalization, by studying some simple architectures, showing that generalization depends on the data distribution as well as the architecture; for example, merely adding a bias to a 2-layer MLP makes generalization impossible for the XOR dataset.

Of course, it is not possible to discuss generalization in deep learning without discussing the results of Zhang et al. (2021), who showed that deep learners can fit with zero training error on random labels using an architecture that generalizes well when fit to the correct labels. Bartlett et al. (2019) and Harvey et al. (2017) found that the VC-dimension for deep ReLU networks is $\mathcal{O}(WL \log W)$ and $\Theta(WU)$ respectively, where $W$ is

the number of parameters, $L$ the number of layers, and $U$ the number of units. Hardt et al. (2016) show that stochastic gradient methods are uniformly stable[8], which implies generalization *in expectation*.

## B  Background

**Definition 1** (Strong convexity). *A function $f : \mathcal{X} \to \mathbb{R}^*$ is $\mu-$strongly convex with respect to $\|\cdot\|$ if $\forall x, y$ in the relative interior of dom $f$ and $\alpha \in (0, 1)$,*

$$f(\alpha x + (1 - \alpha)y) \leq \alpha f(x) + (1 - \alpha)f(y) - \frac{1}{2}\mu\alpha(1 - \alpha)\|x - y\|^2$$

**Definition 2** (Smoothness). *A function $f : \mathcal{X} \to \mathbb{R}$ is $\beta-$smooth with respect to $\|\cdot\|$ if $f \in C^1$ and if $\forall x, y \in dom\ f$,*

$$f(x + y) \leq f(x) + \langle \nabla f(x), y \rangle + \frac{1}{2}\beta\|y\|^2$$

### B.1  The standard covering number guarantee

We reiterate the standard bound under the covering argument from (Duchi (2023)) here. This bounds the deviation of the mini-batch approximation of the supremum from the true value.

Let $f(x) = \sup\|\nabla^2 E(x)\|$ be drawn from some set of functions $\mathcal{F}$, each of whose elements map from $\mathcal{X}$ to $\mathbb{R}$. Define a point-mass empirical distribution on $\{x_i\}_{i=1}^m$ as $P_m = \frac{1}{m}\sum_{i=1}^m \delta_{x_i}$ where $\delta$ is the Dirac delta. For any function $f : \mathcal{X} \to \mathbb{R} \in \mathcal{F}$, let

$$P_m f \triangleq \mathbb{E}_{P_m}[f(X)] = \sum_{i=1}^m f(x_i)$$

be the empirical expectation over a mini-batch and let

$$Pf \triangleq \mathbb{E}_P[f(X)] = \int f(x)dP(x)$$

denote the general expectation under a measure $P$. Suppose the functions in $\mathcal{F}$ are bounded above by $\beta$ (trivially, they are bounded below by 0), and define the metric over $\mathcal{F}$ as $\|f - g\|_\infty = \sup_{x \in \mathcal{X}}|f(x) - g(x)|$. Denote by $N(\delta, \Theta, \rho)$, the covering number for a $\delta-$cover of a set $\Theta$ with respect to a metric $\rho$. Then, we use the standard covering number guarantee (cf. Duchi (2023) Ch. 4) to get

$$P\left(\sup_{f \in \mathcal{F}}|P_m f - Pf| \geq t\right) \leq \exp\left(-\frac{mt^2}{18\beta^2} + \log N(t/3, \mathcal{F}, \|\cdot\|_\infty)\right)$$

## C  Auxiliary Proofs

**Lemma 3.** *Let $f : \mathbb{R}^d \to \mathbb{R}$ be a differentiable function. Then, $\mu-$strong convexity implies:*

  *[(i)]*

  1. *(Polyak-Łojasiewicz (PL) inequality)*

$$\frac{1}{2}\|\nabla f(x)\|^2 \geq \mu(f(x) - f(x^*))$$

  2.

$$\|\nabla f(x) - \nabla f(y)\| \geq \mu\|x - y\|$$

---

[8]An algorithm $A$ is $\epsilon-$uniformly stable if $\forall S, S' \in Z$ for some space $Z$, such that the datasets $S$ and $S'$ differ by at most one example, $\sup_z \mathbb{E}_A[f(A(S); z) - f(A(S'); z)] \leq \epsilon$

*3.*

$$(\nabla f(x) - \nabla f(y))^T (x - y) \leq \frac{1}{\mu} \|\nabla f(x) - \nabla f(y)\|^2$$

*Proof.* [(i)]

Strong convexity implies

$$f(y) \geq f(x) + \nabla f(x)^T (y - x) + \frac{\mu}{2} \|y - x\|^2$$

Minimizing with respect to $y$ yields the result.

2. Strong convexity gives us:

$$(\nabla f(x) - \nabla f(y))^T (x - y) \geq \mu \|x - y\|^2 \tag{4}$$

Applying Cauchy-Schwarz inequality gives us:

$$\|\nabla f(x) - \nabla f(y)\| \|x - y\| \geq (\nabla f(x) - \nabla f(y))^T (x - y)$$
$$\geq \mu \|x - y\|^2$$

where the last step comes from (4).

3. Set $\phi_x(z) = f(z) - \nabla f(x)^T z$. It is easy to see that $\phi_x$ is also $\mu-$strongly convex. Applying the Polyak-Łojasiewicz inequality to $\phi_x(z)$ with $z^* = x$,

$$(f(y) - \nabla f(x)^T y) - (f(x) - \nabla f(x)^T x) = \phi_x(y) - \phi_x(z^*)$$
$$\leq \frac{1}{2\mu} \|\nabla \phi_x(y)\|^2$$
$$\leq \frac{1}{2\mu} \|\nabla f(y) - \nabla f(x)\|^2$$

Swapping $x$ and $y$ in the above,

$$(f(x) - \nabla f(y)^T x) - (f(y) - \nabla f(y)^T y) \leq \frac{1}{2\mu} \|\nabla f(x) - \nabla f(y)\|^2 \tag{5}$$

Adding (5) and (5) yields the result.

$\square$

**Lemma 4.** *If $f : \mathbb{R}^n \to \mathbb{R}$ is smooth and $\mu-$strongly convex, then*

$$\frac{1}{2\mu} \|\nabla f(x)\|^2 \geq f(x) - f(x^*) \geq \frac{\mu}{2} \|x - x^*\|^2$$

*Proof.* The first part is the Polyak-Łojasiewicz inequality. The second part follows from the definition of strong convexity and setting $y = x, x = x^*$ and using $f(x^*) \geq \min_y f(y)$. $\square$

**Theorem 2** (Smooth and strongly convex gradient descent)**.** *Suppose $f : \mathbb{R}^n \to \mathbb{R}$ be $\beta-$smooth and $\mu-$strongly convex. Then with the gradient descent update rule*

$$x_{k+1} = x_k - \frac{1}{\beta} \nabla f(x_k)$$

*where $\frac{1}{\beta}$ is the learning rate, we have*

$$f(x_k) - f(x^*) \leq \left(1 - \frac{\mu}{\beta}\right)^k (f(x_0) - f(x^*))$$

*Consequently, we require $\frac{\beta}{\mu} \log \frac{f(x_0) - f(x^*)}{\epsilon}$ iterations to find an $\epsilon-$optimal point.*

*Proof.* From the above results,

$$f(x_{k+1}) \leq f(x_k) - \frac{1}{2\beta}\|\nabla f(x_k)\|^2$$

and Lemma 4 gives us

$$\|\nabla f(x_k)\|^2 \geq 2\mu(f(x_k) - f(x^*))$$

Therefore,

$$\begin{aligned}
f(x_{k+1}) - f(x^*) &\leq f(x_k) - f(x^*) - \frac{1}{2\beta}\|\nabla f(x_k)\|^2 \\
&\leq f(x_k) - f(x^*) - \frac{\mu}{\beta}(f(x_k) - f(x^*)) \\
&= \left(1 - \frac{\mu}{\beta}\right)(f(x_k) - f(x^*)) \\
f(x_k) - f(x^*) &\leq \left(1 - \frac{\mu}{\beta}\right)^k (f(x_k) - f(x^*)) \\
&\leq \exp\left(-\frac{\mu k}{\beta}\right)(f(x_0) - f(x^*))
\end{aligned}$$

Therefore we need $k = \frac{\beta}{\mu}\log\frac{f(x_0)-f(x^*)}{\epsilon}$ iterations for $\epsilon-$optimality. $\qquad\square$

Note that strong convexity guarantees optimality. $\beta-$smoothness can only assure $\epsilon-$criticality. This implies the existence of global minima.

Strong convexity is a necessary condition for learnability, as along with $\beta-$smoothness, it can be shown that such problems are learnable (Shalev-Shwartz and Ben-David (2014)). Additionally, strong convexity provides a quadratic lower bound on the growth of the loss function, which implies that the convexity condition will never be violated in the local domain of the function in the context of deep regression with regularization.

## D   Additional details on experimental setup

Table 3 shows the hyper-parameter space for our experiments on the image and OpenML datasets. For our Bayesmark experiments, we use the default hyper-parameter space.

## E   Additional loss landscapes

This section shows examples of loss landscapes with low and high strong convexity for more datasets (as done in Figure 1). For brevity, we use `btsc` to mean the `blood-transfusion-service-center` dataset.

## F   On using one epoch as a heuristic

R6ZG2:W7 In this section, we show additional experimental verification that using the sharpness after one epoch is a reasonable choice (and more informative than the validation loss). Figure 5 shows plots of our sharpness measure and the validation loss over epochs for the HPOBench and image datasets.

Note that the goal of computing an estimate of the sharpness after one epoch is to obtain an upper bound on the sharpness across the entire space. Therefore, we expect that the sharpness $\zeta$, if expressed as a function of epochs $t$ (so that $\zeta(t)$ is the sharpness measured after epoch $t$), has the property that for some $t_0$, $\zeta(t) \leq \zeta(1)\forall t > t_0$.

This trend is observed in the plots of Figure 5. Across all plots, the red line indicates the validation loss, and the green line indicates the sharpness. With the exception of the vehicle, blood-transfusion-service-center, and segment datasets, all datasets show the property we described above. Moreover, we note that we *expect*

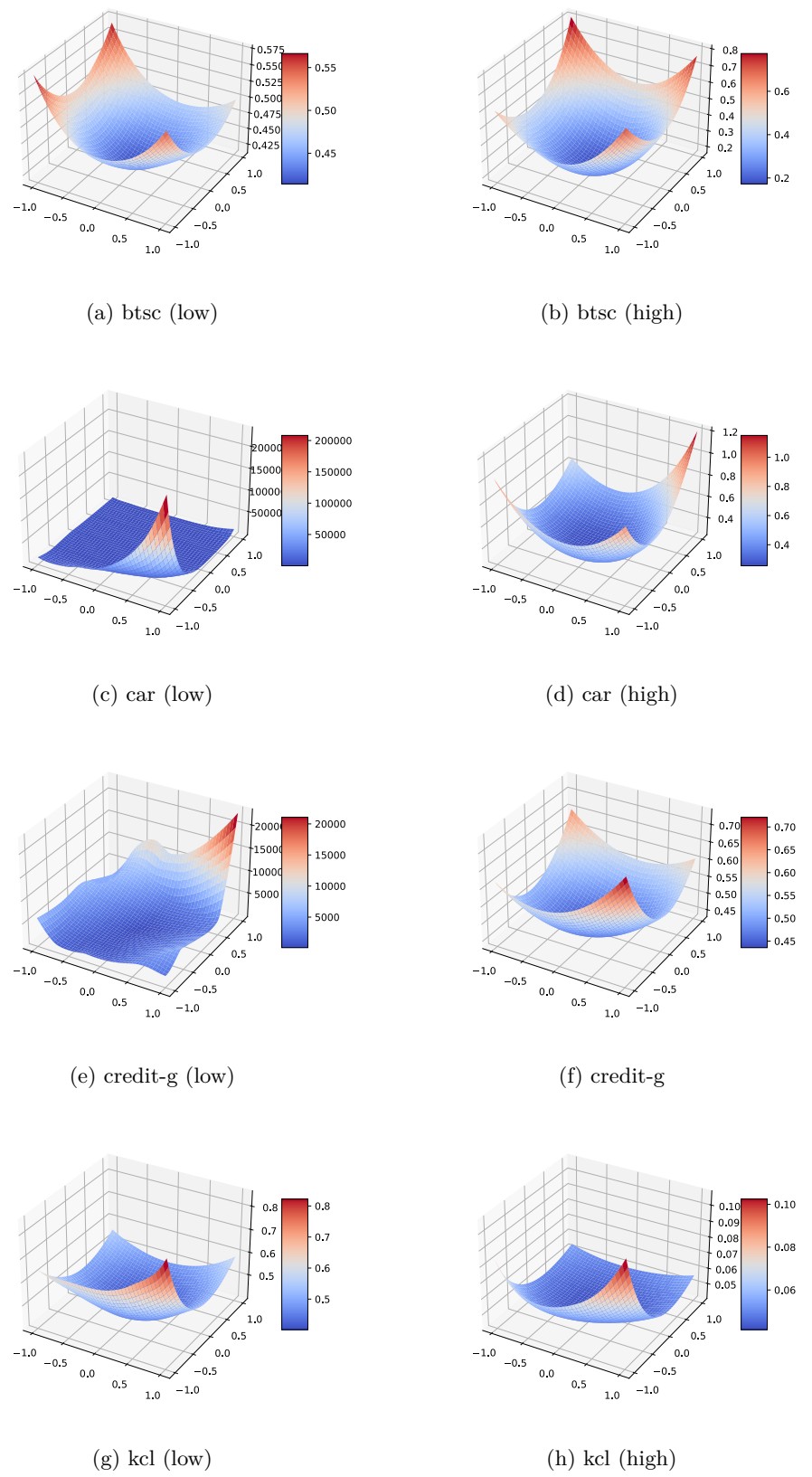

(a) btsc (low)

(b) btsc (high)

(c) car (low)

(d) car (high)

(e) credit-g (low)

(f) credit-g

(g) kcl (low)

(h) kcl (high)

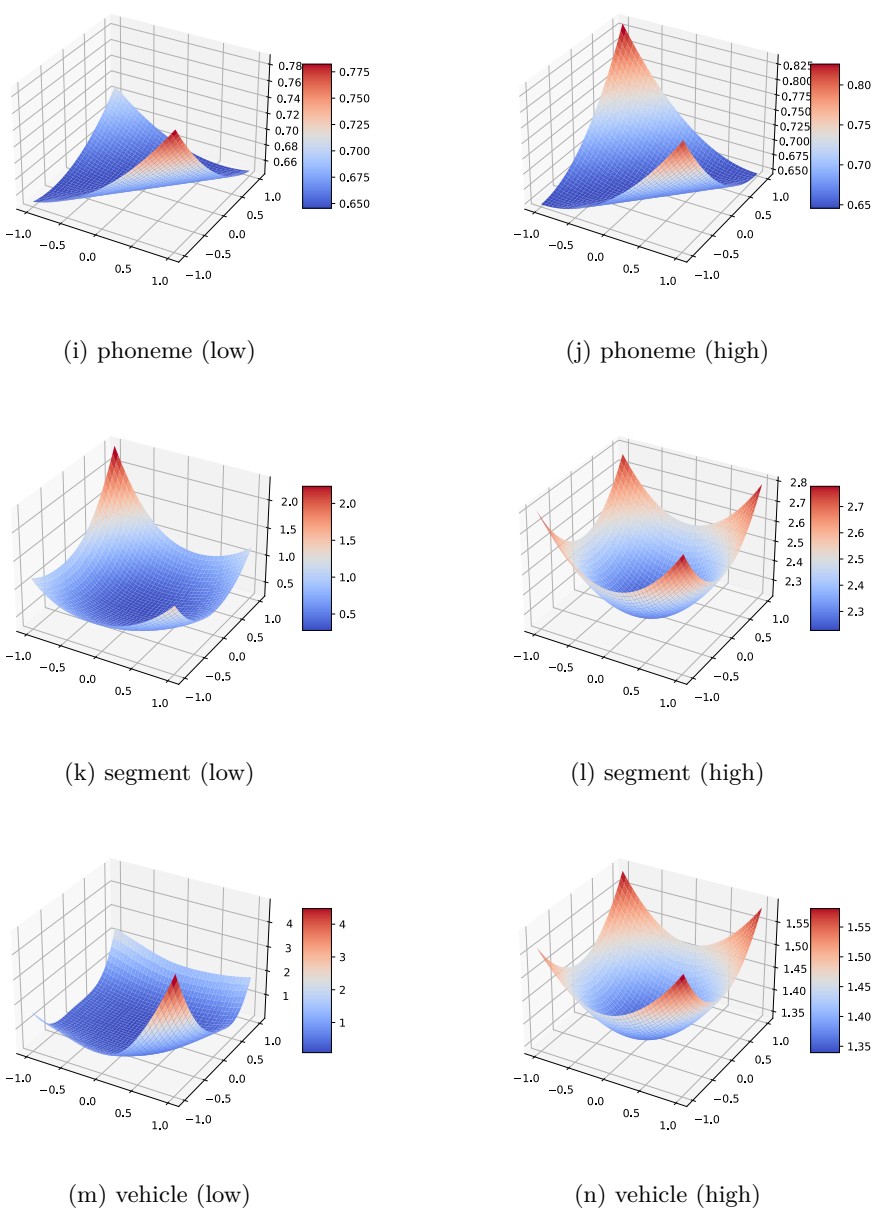

(i) phoneme (low)

(j) phoneme (high)

(k) segment (low)

(l) segment (high)

(m) vehicle (low)

(n) vehicle (high)

Figure 4: More loss landscapes with low and high strong convexity values.

Table 3: Hyper-parameters used in this study. Ranges are inclusive. Unless specified, the ranges are linear. For Bayesmark, we use the default hyper-parameter set, whose size is 134M.

| OpenML | |
|---|---|
| **Hyper-parameter** | **Range** |
| Network depth | $(1, 4)$ |
| Network width | $(16, 1024)$, $\log_2$ |
| Batch size | $(4, 256)$, $\log_2$ |
| Initial learning rate | $(10^{-5}, 1.0)$, $\log_{10}$ |
| MNIST | |
| **Hyper-parameter** | **Range** |
| Number of filters | $(2, 6)$ |
| Kernel size | $(2, 6)$ |
| Padding | Valid, same |
| Number of conv blocks | $(1, 3)$ |
| SVHN | |
| **Hyper-parameter** | **Range** |
| Number of filters | $(2, 6)$ |
| Kernel size | $(2, 6)$ |
| Padding | Valid, same |
| Number of conv blocks | $(1, 3)$ |
| Dropout rate | $(0.2, 0.5)$ |
| Final dropout rate | $(0.2, 0.5)$ |
| Number of units | $(32, 512)$, $\log_2$ |

Table 4: Details of configurations and performance for landscapes of Figure 4. BS = batch size; $\alpha = L_2$ regularization factor. Acc and AUC show training accuracy and AUC, and generalization error in parentheses. Datasets where the sharper landscapes outperformed the flatter minima are in red.

| **Dataset** | **Depth** | **Width** | **BS** | $\alpha$ | **LR** | $\mu$ | **Acc** | **AUC** |
|---|---|---|---|---|---|---|---|---|
| btsc | 3 | 128 | 4 | $10^{-7}$ | $10^{-5}$ | 0.57 | 88.71 (10.05) | 91.55 (12.90) |
| | 3 | 1024 | 4 | $10^{-5}$ | $10^{-5}$ | 1.08 | 93.18 (21.56) | 98.10 (39.84) |
| car | 4 | 16 | 256 | $10^{-8}$ | 1.0 | 0.16 | 70.03 (0.09) | 85.70 (-0.43) |
| | 1 | 32 | 16 | $10^{-6}$ | $10^{-4}$ | 1.09 | 94.22 (1.78) | 99.68 (0.16) |
| credit-g | 2 | 32 | 32 | $10^{-5}$ | 1.0 | 0.03 | 70 (0) | 50.92 (0.92) |
| | 1 | 128 | 4 | $10^{-3}$ | $10^{-4}$ | 0.78 | 100 (32) | 100 (31.57) |
| kcl | 4 | 32 | 32 | $10^{-3}$ | $10^{-5}$ | 0.13 | 96.10 (9.84) | 95.79 (13.18) |
| | 1 | 128 | 4 | $10^{-6}$ | $10^{-4}$ | 1.79 | 98.58 (15.25) | 99.59 (21.69) |
| phoneme | 1 | 16 | 8 | 0.1 | $10^{-5}$ | 0.08 | 100 (26.99) | 100 (34.55) |
| | 1 | 64 | 4 | 0.1 | $10^{-3}$ | 0.97 | 100 (28.89) | 100 (45.23) |
| segment | 3 | 128 | 4 | $10^{-3}$ | $10^{-5}$ | 0.03 | 99.42 (23.66) | 100 (4.85) |
| | 3 | 256 | 16 | 1.0 | $10^{-5}$ | 0.08 | 99.66 (27.37) | 100 (6.02) |
| vehicle | 2 | 16 | 8 | $10^{-4}$ | $10^{-4}$ | 0.11 | 100 (40) | 100 (17.13) |
| | 1 | 16 | 64 | 0.01 | $10^{-4}$ | 0.33 | 92.52 (36.57) | 98.63 (12.04) |

the validation loss to go down over time, and as such it is not informative as a metric for choosing one hyper-parameter configuration over another.

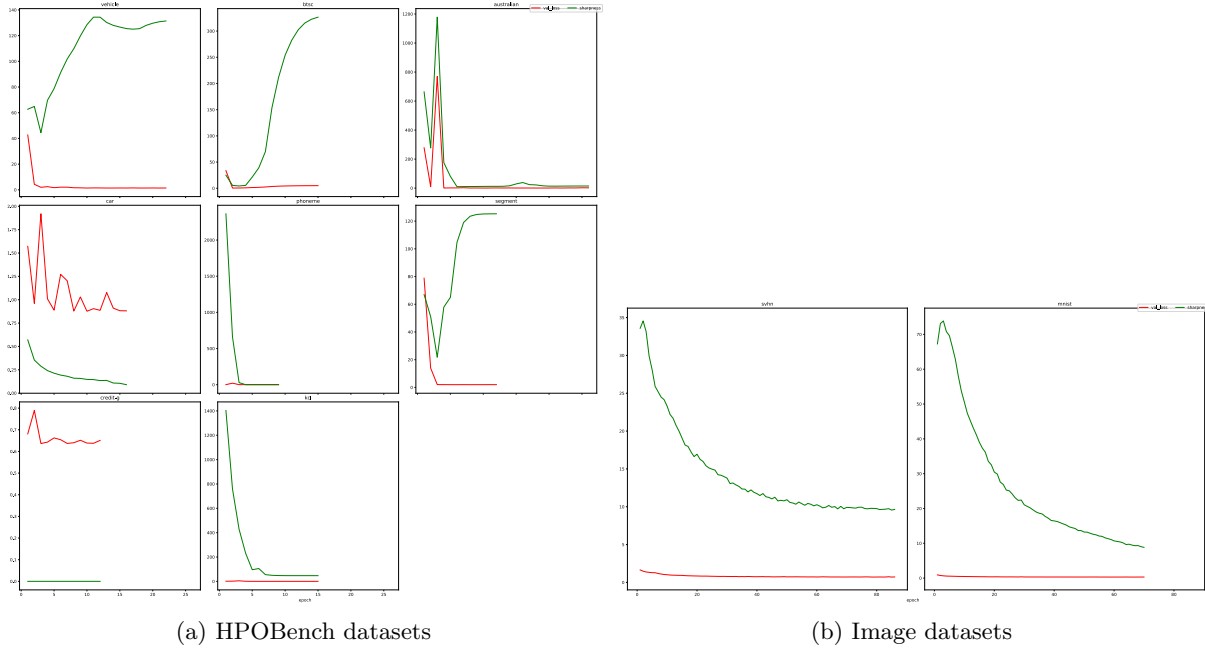

(a) HPOBench datasets

(b) Image datasets

Figure 5: Progression of validation loss and sharpness over epochs on HPOBench and image datasets

Moreover, we found that the IoU metric between the 10 lowest validation errors after one epoch with those after 10 epochs (the same experimental setup used in Figure 3) was significantly lower. This makes sense, since hyper-parameters influence training dynamics over epochs as well, which leads to validation loss being an unreliable predictor for future performance.

