# OpenReview forum: "Flatness-guided hyper-parameter optimization"
_TMLR — Rejected by TMLR_

### Review · Reviewer_6ZG2 · 2024-07-15

**Summary Of Contributions:**

This paper proposes a way of choosing hyper-parameters. In its experiments, it does this:

* Randomly draw 50 hyper-parameter configurations;
* Train 1 epoch for each of the 50 configurations;
* Select the top 10 runs based on a flatness measure;
* Continue training the 10 configurations to the end, and take the best.

The authors evaluated this method on 14 classification datasets. The datasets are all very small, of the MNIST level.

**Audience:**

Yes

**Broader Impact Concerns:**

None.

**Claims And Evidence:**

No

**Requested Changes:**

Please rewrite the paper and address the points I raised above.

**Strengths And Weaknesses:**

The technical correctness of this paper is quite suspicious to me.

My first question is about this experimental setting: Instead of the proposed flatness measure, why not just use the validation loss? In order to show that the proposed flatness measure makes sense, **you not only have to show that the final validation loss is correlated with the flatness measure, but also you have to show it is _stronger_ than the correlation between the final validation loss and the validation loss after the first epoch**.

Currently, the authors do not seem to have done such experiments. Instead, they compare with other hyper-parameter optimization methods; these baseline methods are briefly introduced in Related Work, but the Experiment section lacks precise description of what exactly these baseline methods do, and what are the detailed experimental settings of these baselines. So it is difficult to judge if the comparison is fairly conducted.

Moreover, there are discrepancies in the methods being compared: in Table 1, they are  FGHPO (proposed), Hyperopt, Random, TuRBO, HEBO; in Table 2 they are FGHPO, TPE, Random, TuRBO, HEBO, BOHB (so Hyperopt disappeared, TPE and BOHB newly appeared); in Figure 2 they are AHSC, TuRBO, HEBO, Random, Hyperopt, BOHB (so the proposed FGHPO disappeared (!), and AHSC newly appeared). The paper does not explain these discrepancies.

There are other places in the paper I found questionable:

* On page 4, the authors denote the loss by $E(x;H)$, where $x$ is the data point and $H$ is the hyper-parameter (!). Usually we would denote the loss by $E(x;\theta)$ where $\theta$ is the *model parameter*; Optimization hyper-parameters, such as learning-rate, won't affect the loss; Model hyper-parameters, such as hidden dimension, will affect the loss function but, how would you compare the Hessian $\nabla^2 E$ between different model sizes?

* In the beginning of Section 3.2, where the authors define the proposed FGHPO, they use the whole Hessian $\nabla^2 E$. Then, they use the symbol $\nabla_W^2 E$, where $W$ represents the weights of a neural network. So I assume $\nabla_W^2 E$ is the same as $\nabla^2 E$, but what does $\nabla_W^2 E$ even mean? Because $\nabla^2 E$ is a rank-2 tensor; if you want to write it more explicitly, you should write $\nabla_{W,W}^2 E$.

* On page 5, it says "Below, we show that the strong convexity... is given by"
$$
\inf\lVert\nabla^2 E\rVert\propto\frac{1}{m}\inf\frac{\lVert a_j^{[L-1]}\rVert}{\lVert W^{[L]}\rVert}
$$
but this is not proven by Theorem 1 below. Theorem 1 claims that
$$
\lVert\nabla_W^2 E\rVert\geq\frac{\lVert a_j^{[L-1]}\rVert}{\lVert W^{[L]}\rVert},
$$
but it proves that
$$
\lVert\nabla_{W^{[L]}}^2 E\rVert\geq\frac{\lVert a_j^{[L-1]}\rVert}{\lVert W^{[L]}\rVert}.
$$
The three formulas are not equivalent; and I don't feel like I'm ready to check the correctness of the proof. (Here, I assume $\nabla_{W^{[L]}}^2 E$ means $\nabla_{W^{[L]},W^{[L]}}^2 E$, which is what the authors do: they differentiate $E$ by $W^{[L]}$ twice.)

* So, how exactly do you calculate $\inf\lVert\nabla^2 f(x;h)\rVert$ in Algorithm 1? (and why $E$ becomes $f$ here?...)

---

> ### Author Response · Authors · 2024-07-25
> **Rebuttal to Reviewer 6ZG2**
>
> Thank you for your review. We have responded to your comments below.
>
> **W1:** In order to show that the proposed flatness measure makes sense, **you not only have to show that the final validation loss is correlated with the flatness measure, but also you have to show it is *stronger* than the correlation between the final validation loss and the validation loss after the first epoch**.
>
> Thank you for the suggestion. We have run this test and added the result to our paper (see `R6ZG2:W1` on page 10).
>
> **W2:** [...] these baseline methods are briefly introduced in Related Work, but the Experiment section lacks precise description of what exactly these baseline methods do, and what are the detailed experimental settings of these baselines.
>
> As stated in Section 2, the related work section in the main text was intentionally brief; a more detailed description of the methods we baseline against is in Appendix A.  We use the default settings for each baseline (see `R6ZG2:W2` on page 9).
>
> **W3:** Moreover, there are discrepancies in the methods being compared: in Table 1, they are FGHPO (proposed), Hyperopt, Random, TuRBO, HEBO; in Table 2 they are FGHPO, TPE, Random, TuRBO, HEBO, BOHB (so Hyperopt disappeared, TPE and BOHB newly appeared); in Figure 2 they are AHSC, TuRBO, HEBO, Random, Hyperopt, BOHB (so the proposed FGHPO disappeared (!), and AHSC newly appeared). The paper does not explain these discrepancies.
>
> Thank you for pointing this out. Hyperopt is an implementation of TPE, see [1]. We have updated this now (see `R6ZG2:W3a` on page 10). BOHB did not newly appear in Table 2, it is also present in Table 1. We apologize for the oversight in Figure 2; AHSC was an earlier name for our method. We have changed this in the updated version.
>
> **W4a:** Usually we would denote the loss by $E(x; \theta)$ where $\theta$ is the *model parameter.*
>
> Thank you for pointing out that we did not specify this in the notation section. We have now updated this (see `R6ZRG2:W4a` on page 4). It is important to note that this paper focuses on the role of hyper-parameters in the sharpness of the loss landscapes (see W4c below), so we parameterize using them instead.
>
> **W4b:** Optimization hyper-parameters, such as learning-rate, won't affect the loss;
>
> While true, learning rate (and batch size) do affect the SGD dynamics and influence the width of the minima it finds, see [2].
>
> **W4c:** Model hyper-parameters, such as hidden dimension, will affect the loss function but, how would you compare the Hessian $\nabla^2 E$ between different model sizes?
>
> Note that the loss function is a function of the data, the weights, and the hyper-parameters (the latter materialize a specific neural network). When the (training) dataset is fixed, the weights influence the loss, and the hyper-parameters, by materializing different subsets of possible loss surfaces, also influence the sharpness of the loss observed. Because the loss function itself is the same (cross-entropy), the sharpness can be computed across different values of hyper-parameters. Our paper explains this connection in Section 1, page 2. This is the same reason we parameterize the loss by the hyper-parameters, although it is also influenced by the specific parameters.
>
> **W5:** So I assume $\nabla^2_W E$ is the same as $\nabla^2 E$, but what does $\nabla_W^2 E$ even mean? Because $\nabla^2 E$ is a rank-2 tensor; if you want to write it more explicitly, you should write $\nabla^2_{W, W} E$.
>
> Your assumption is correct; however, $\nabla^2 E$ is a **rank-4** tensor since $W$ is 2D. For the sake of brevity, we do not use the more explicit notation, since the term Hessian is typically assumed to mean the gradient is taken twice with respect to the weights. We have clarified this in the notation section (see `R6ZG2:W5` on page 4).
>
> **W6:** [...] These formulas are not equivalent
>
> You are right, we apologize for the oversight. We have now clarified that we compute a *lower bound*, but that it cannot be guaranteed to be the *greatest* lower bound (see `R6ZG2:W6a` on page 4). We therefore introduce the notation $\text{lb } f$ to denote a lower bound and use this instead (see `R6ZG2:W6b` on page 8).
>
> **W7:** How exactly do you calculate $\inf \lVert \nabla^2 f(x; h) \rVert$ in Algorithm 1?
>
> The lower bound on the Hessian norm is computed using the results of Theorem 1, which we clarified in Section 3.2.
>
> ### References
>
> [1] Bergstra, J., Komer, B., Eliasmith, C., Yamins, D., & Cox, D. D. (2015). Hyperopt: a python library for model selection and hyperparameter optimization. *Computational Science & Discovery*, *8*(1), 014008.
>
> [2] Jastrzębski, S., Kenton, Z., Arpit, D., Ballas, N., Fischer, A., Bengio, Y., & Storkey, A. (2017). Three factors influencing minima in sgd. *arXiv preprint arXiv:1711.04623*.

---

> > ### Comment · Reviewer_6ZG2 · 2024-08-13
> > **Thanks for the additional experiments**
> >
> > Unfortunately, I believe this work is seriously flawed.
> >
> > The main story consists of two steps: (1) the authors claim that the Frobenius norm of the per-batch Hessian, observed at the point that the model is trained for one epoch, is an indicator for predicting the final validation loss; (2) the authors prove a "lower bound" for the Frobenius norm, which is given by $\lVert a_j^{[L-1]}\rVert / \lVert W^{[L]}\rVert$ where $W^{[L]}$ is the weight matrix of the last layer and $a_j^{[L-1]}$ is the activation before the last layer; then, the authors use $\lVert a_j^{[L-1]}\rVert / \lVert W^{[L]}\rVert$ as an indicator for predicting the final validation loss.
> >
> > I won't comment on the first step because it's essentially irrelevant: even the proof for the proposed lower bound is correct (I haven't checked that), relations regarding the magnitude of $\lVert a_j^{[L-1]}\rVert / \lVert W^{[L]}\rVert$ won't imply any information regarding the magnitude of the Frobenius norm of the Hessian. Basically the authors are just calculating the norm of a very small portion of the large Hessian tensor, which is like using the absolute value of the (0,0)-entry as a lower bound for a large NxN matrix -- and from the authors' response to W5 above, they seem totally comfortable with ignoring all the (i,j)-entries for i≠j -- I don't see why one can believe this bold calculation still leads to any meaningful conclusion regarding the Hessian or flatness.
> >
> > So, directly regarding the second step: I believe it's seriously flawed because the quantity $\lVert a_j^{[L-1]}\rVert / \lVert W^{[L]}\rVert$ is not comparable without controlling for the hyper-parameters. The norm $\lVert W^{[L]}\rVert$ depends on the size of the matrix $W$; and the norm value at the point that the model is trained for one epoch, depends on the learning-rate and the initialization. From the authors' response to W4c above, they seem to believe that the quantity $\lVert a_j^{[L-1]}\rVert / \lVert W^{[L]}\rVert$ can be directly compared between different hyper-parameter settings "just like the loss"; unfortunately this is not true: because by scaling up $W^{[L]}$ and scaling down $W^{[L-1]}$ one can arbitrarily decrease $\lVert a_j^{[L-1]}\rVert / \lVert W^{[L]}\rVert$ (and vice versa) without changing the final loss function at all! Therefore, I don't believe the value of $\lVert a_j^{[L-1]}\rVert / \lVert W^{[L]}\rVert$ can indicate anything, because it is arbitrary.
> >
> > Specific to the additional experiments in Appendix F, here are my comments:
> >
> > 1. I don't think you get my point; instead of showing the validation loss of one run, you have to conduct multiple runs and take the set of validation loss values at epoch 1, 2, 3, ... and calculate its correlation with the set of validation loss values at the final epoch.
> >
> > 2. What is the unit for y-axes in Figure 5?
> >
> > 3. "This means that we expect the sharpness (as measured by our equation) to go down over time" -- I don't think you have mathematically proved this claim; why do you expect the sharpness measure to go down over time? And what is your explanation for the 3 cases that it does not go down (as in "vehicle", "btsc" and "segement" where the green line blows up)?

---

> ### Comment · Reviewer_6ZG2 · 2024-08-02
> **Thanks for the response**
>
> Thanks for answering my questions and making the changes. I'm still not convinced yet; before we discuss more details about the theory and experiments, can you also show a graph of how the proposed flatness measure changes during training, together with the validation loss? Eventually, the validation loss will have correlation 1 with the final validation loss (because they are the same thing); so it will overtake the correlation score between flatness and final validation at some point -- when does that happen? And how does that change across different tasks and different random initializations? This is an important experiment to justify your choice of training one epoch in the proposed method.

---

> > ### Author Response · Authors · 2024-08-08
> > **On the one epoch heuristic**
> >
> > Thank you for your questions! We have added the graph for the HPOBench and image datasets in a new revision, please see the new Appendix F. Rather than correlation (which would not make sense between a series of points and a single, final validation), we instead show that the goal of using the sharpness estimate after one epoch was to obtain an upper bound on the sharpness across the landscape. This means that we expect the sharpness (as measured by our equation) to go down over time, which it does for most of the datasets.

---

### Review · Reviewer_uHZU · 2024-07-19

**Summary Of Contributions:**

This work examines a computationally efficient hyperparameter optimization scheme. The criterion for optimality is the flatness of the problems loss after one epoch. The flatness is computed via the frobenius norm of the Hessian. The scheme is tested on 14 datasets.

**Audience:**

Yes

**Broader Impact Concerns:**

no broader impact statement needed

**Claims And Evidence:**

No

**Requested Changes:**

- Fig. 1: Why is the accuracy negative? The wording should be more precise. Is this is loss landscape in terms of the hyper parameters? What are the axes actually? Or is it the loss landscpe of the weights of the model? How did you compute the convexity?

- algorithm 1, line 6: The connection between generalization and flatness was made at a given minimum, and not throughout training dynamics. It seems unreasonable to assume any training would have converged after an epoch, and therefore give a good idea of the properties locally around the minimum. Is this not just going to favour slower learning rates?

- Fig 2: Averge runtimes? Examples?  Respecify the computational environment.

- You say “since it is not necessary for a landscape to have flat minima to generalize well” — this should be discussed at an earlier point already. The fact that there is an ongoing (controvercial) discussion about the connection between flatness and generalization should be highlighted earlier in the related work.

- Lemma 1’s proof contains an inequality. The statement does not.

- Section 5 conclusions and future work reads more like “Limitations”.

- I do not have sufficient experience in hyperparameter optimization to judge the baselines used in this work. The very basic one is running a train/test split and selecting the hyperparameters based on the test error achieved after a given time. It would be useful to see this, or at least be reminded in the experimtal section what classes of algorithms the acronyms belong to.

- Figure 3: Over the first ten epochs of what? Configurations of what?

**Strengths And Weaknesses:**

Strengths
- The idea is interesting.

Weaknesses
- The paper is poorly structured, the method section itself lacks an overview, theorems and lemma are stated without much context.
- The content shows inconsistencies even on a very superficial level (e.g. Fig.1: negative accuracy, Lemma 1: States an equality, but proof contains an inequality). I list more in the requested changes.

---

> ### Author Response · Authors · 2024-07-25
> **Rebuttal to Reviewer uHZU**
>
> Thank you for reviewing our paper. Below, we respond to the points you raised:
>
> **W1:** The paper is poorly structured, the method section itself lacks an overview, theorems and lemma are stated without much context.
>
> We agree. We have added more context to the theoretical results:
>
> - We discuss why it was necessary to show that the lower bound on the Hessian norm is relevant at all (see `RuHZU:W1a` on page 4).
> - We added a subsection (Section 3.3) for our theoretical results. We also clarified the purpose of the theorem (see `RuHZU:W1b` on page 6) and the two lemmas preceding it (see `RuHZU:W1c` on page 6).
>
> **W2:** Fig. 1: Why is the accuracy negative? The wording should be more precise. Is this is loss landscape in terms of the hyper parameters? What are the axes actually? Or is it the loss landscpe of the weights of the model? How did you compute the convexity?
>
> We apologize for the confusion. Figure 1 (the part showing negative values) shows the *generalization errors* for AUC and accuracy, computed as $\text{acc}_{\text{train}} - \text{acc}_{\text{test}}$. We have now clarified this (see `RuHZU:W2a` on page 2).
>
> We use the method of Li et al. (2018) as mentioned in the caption to visualize the loss landscape in terms of the weights of the neural network. Briefly, the two axes are random direction vectors sampled from a filter-normalized random Gaussian distribution (feedforward layers are treated as convolutional layers with 1x1 feature maps). The filter-wise normalization allows comparison between different loss functions. We apologize for the wording confusion: the "strong convexity" is merely the lower bound on the Hessian norm, computed using the result of Theorem 1. We have clarified this in the updated caption.
>
> **W3:** algorithm 1, line 6: The connection between generalization and flatness was made at a given minimum, and not throughout training dynamics. It seems unreasonable to assume any training would have converged after an epoch, and therefore give a good idea of the properties locally around the minimum. Is this not just going to favour slower learning rates?
>
> It is true that the models do not converge after one epoch; however, the exact sharpness values are not as important as the relative sharpness, since out of the 50 initial configurations, we only fully train the 10 with the lowest sharpness. As we discussed in Section 4.2, this set of 10 configurations does change over epochs, and we argue that this is an exploration-exploitation trade-off.
>
> Regarding your question about favoring slower learning rates, we believe you are referencing the results of [1], which do not consider the influence of other hyper-parameters simultaneously being explored. As such, the method does not necessarily favor low learning rates.
>
> **W4:** Fig 2: Averge runtimes? Examples? Respecify the computational environment.
>
> We are currently running experiments on this and will add another comment when the results are ready.
>
> **W5:** You say “since it is not necessary for a landscape to have flat minima to generalize well” — this should be discussed at an earlier point already. The fact that there is an ongoing (controvercial) discussion about the connection between flatness and generalization should be highlighted earlier in the related work.
>
> Thank you for the suggestion. We have added this to the Related Work section (see `RuHZU:W5` on page 3).
>
> **W6:** Lemma 1’s proof contains an inequality. The statement does not.
>
> You are right. We have made this more consistent. Moreover, based on feedback from Reviewer `6ZG2`, we have improved the notation on this (see `R6ZG2:W6a` on page 4 and `R6ZG2:W6b` on page 8).
>
> **W7:** Section 5 conclusions and future work reads more like “Limitations”.
>
> We agree. We have renamed this section.
>
> **W8:** The very basic one is running a train/test split and selecting the hyperparameters based on the test error achieved after a given time. It would be useful to see this, or at least be reminded in the experimtal section what classes of algorithms the acronyms belong to.
>
> We have now added more discussion on these points (see `RuHZU:W8a` on page 10 and `RuHZU:W8b` on page 9).
>
> **W9:** Figure 3: Over the first ten epochs of what? Configurations of what?
>
> Thank you for pointing out the confusion. We have revised this caption (see `RuHZU:W9` on page 11).
>
> ### References
>
> [1] Jastrzębski, S., Kenton, Z., Arpit, D., Ballas, N., Fischer, A., Bengio, Y., & Storkey, A. (2017). Three factors influencing minima in sgd. *arXiv preprint arXiv:1711.04623*.

---

> ### Author Response · Authors · 2024-08-08
> **A gentle reminder**
>
> Dear Reviewer uHZU,
>
> Thank you for your review. We wanted to gently remind you and ask if you had further questions or suggestions based on our revised text and rebuttal.
>
> - Authors

---

### Review · Reviewer_bkEw · 2024-07-21

**Summary Of Contributions:**

The paper proposes a white-box approach to hyper-parameter optimization (HPO) by focusing on improving the flatness of the loss to enhance generalization. The authors establish a relationship between the Hessian norm and sharpness metrics and use this to derive semi-empirical estimates for the sharpness of the loss. Their method seeks to find hyper-parameter configurations that minimize this sharpness, aiming for better performance at reduced computational costs.

**Audience:**

Yes

**Broader Impact Concerns:**

I have no concerns regarding the broader impact of this paper.

**Claims And Evidence:**

No

**Requested Changes:**

Please address the weaknesses mentioned above.

**Strengths And Weaknesses:**

## Strengths

1. The idea of using flatness as a guide for hyper-parameter optimization is innovative and aligns with recent research linking flat minima to better generalization.

2. The method is validated across multiple datasets, showing that it can achieve competitive performance with reduced computational costs.

## Weaknesses

1. The claimed contributions are not clearly delineated, making the paper difficult to follow. It is hard to understand what the main novel contributions are and how they significantly advance the field.

2. The overall purpose and goals of the paper are not clearly stated, which makes it challenging to grasp the significance and direction of the research.

3. The importance and implications of the theoretical results, particularly Theorem 1, are not well-explained. It is difficult to understand the motivations for establishing Theorem 1 and its relationship to the main purpose of the paper.

4. The method section lacks sufficient detail, particularly regarding the development and motivation of the proposed algorithm.

---

> ### Author Response · Authors · 2024-07-25
> **Rebuttal to Reviewer bkEw**
>
> Thank you for reviewing our paper. We address the weaknesses below:
>
> **W1:** The claimed contributions are not clearly delineated, making the paper difficult to follow. It is hard to understand what the main novel contributions are and how they significantly advance the field.
>
> We agree; we have elaborated on our contributions section (see `RbkEw:W1a` on page 2, and  `RbkEw:W1b` and `RbkEw:W1c` on page 3).
>
> **W2:** The overall purpose and goals of the paper are not clearly stated, which makes it challenging to grasp the significance and direction of the research.
>
> We have added more text on the specific direction and goals of this research in the introduction (see `RbkEw:W2` on page 2).
>
> **W3:** The importance and implications of the theoretical results, particularly Theorem 1, are not well-explained. It is difficult to understand the motivations for establishing Theorem 1 and its relationship to the main purpose of the paper.
>
> Thank you for the suggestion. We have also received a similar concern from Reviewer `uHZU` and have made changes to the paper:
>
> - We explained better the motivation for the proofs in our paper. For example, we added text on why it was necessary to relate the lower bound on the Hessian norm (which we use) to the sharpness measures used in literature (see `RuHZU:W1a` on page 4).
> - We explicitly state now that the result of Theorem 1 is the expression used in our algorithm to estimate the sharpness (see `RuHZU:W1b` on page 6).
> - We note that the lemmas are auxiliary results that are used in the proof for Theorem 1 (see `RuHZU:W1c` on page 6).
>
> **W4:** The method section lacks sufficient detail, particularly regarding the development and motivation of the proposed algorithm.
>
> You are right, we have now moved the paragraph that motivates our approach to the beginning of the Method section, which is more appropriate. Further, we start with the *sharpness* as the motivation, and then move to the lower bound on the Hessian norm, which we use to estimate it (see `RbkEw:W4` and `RuHZU:W1a` on page 4).

---

> ### Author Response · Authors · 2024-08-08
> **Gentle reminder**
>
> Dear Reviewer bkEw,
>
> Thank you for providing feedback on our paper. Could you please let us know if you have further concerns about our paper based on the revisions and rebuttal?
>
> - Authors

---

### Decision · Action_Editor_DWhT · 2024-10-02

**Recommendation:** Reject

**Comment:**

Some main points raised in the paper are not well supported.

(i) The paper uses the ratio of the activation before the last layer to the weight of the last layer as an indicator for predicting the final validation loss. However, the effectiveness of this criterion is not fully justified.

(ii) The generalization of the proposed method to other datasets is not thoroughly explored.

These concerns were pointed out by the reviewers but were not adequately addressed by the authors. Therefore, the paper is not ready for publication yet.

**Audience:**

Yes.

**Claims And Evidence:**

Some claims about the effectiveness of the proposed algorithm are not fully supported by their theory and experiments. More details are provided in the comments below.